Detection of single nucleotide polymorphisms in virus genomes assembled from high-throughput sequencing data: large-scale performance testing of sequence analysis strategies

Rollin Johan 1
Bester Rachelle 2 3
Brostaux Yves 4
Caglayan Kadriye 5
De Jonghe Kris 6
Eichmeier Ales 7
Foucart Yoika 6
Haegeman Annelies 6
Koloniuk Igor 8
Kominek Petr 9
Maree Hans 2 3
Onder Serkan 10
Posada Céspedes Susana 11 12
Roumi Vahid 13
http://orcid.org/0000-0002-9677-0530 Šafářová Dana 14
Schumpp Olivier 15
Ulubas Serce Cigdem 16
Sõmera Merike 17
Tamisier Lucie 18 19
Vainio Eeva 20
van der Vlugt Rene AA 21
Massart Sebastien 1 sebastien.massart@uliege.be
1 Laboratory of Plant Pathology—TERRA—Gembloux Agro-Bio Tech, University of Liège , Gembloux , Belgium
2 Citrus Research International , Matieland , South Africa
3 Department of Genetics, Stellenbosch University , Matieland , South Africa
4 Laboratory of Statistics, Computer Science and Modelling Applied to Bioengineering, TERRA, Gembloux Agro-Bio Tech, Teaching and Research Centre, University of Liège , Gembloux , Belgium
5 Plant Protection Department, Agricultural Faculty, Hatay Mustafa Kemal University , Hatay , Turkey
6 Fisheries and Food (ILVO), Plant Sciences Unit, Flanders Research Institute for Agriculture , Merelbeke , Belgium
7 Mendeleum—Institute of Genetics, Faculty of Horticulture, Mendel University in Brno , Lednice , Czech Republic
8 Biology Centre CAS , Ceske Budejovice , Czech Republic
9 Crop Research Institute , Praha , Czech Republic
10 Department of Plant Protection, Faculty of Agriculture, Eskişehir Osmangazi University , Eskişehir , Turkey
11 Department of Biosystems Science and Engineering, ETH Zurich , Basel, 4058 , Switzerland
12 Swiss Institute of Bioinformatics (SIB) , Basel , Switzerland
13 Plant Protection Department, Faculty of Agriculture, University of Maragheh , Maragheh , Iran
14 Department of Cell Biology and Genetics, Faculty of Science, Palacký University Olomouc , Olomouc , Czech Republic
15 Plant Protection Department, Agroscope , Nyon , Switzerland
16 Plant Production and Technologies Department, Ayhan Şahenk Faculty of Agricultural Science and Technologies, Niğde Ömer Halisdemir University , Niğde , Turkey
17 Department of Chemistry and Biotechnology, Tallinn University of Technology , Tallinn , Estonia
18 Pathologie Végétale, Institut National de la Recherche pour l’Agriculture, l’Alimentation et l’Environnement (INRAE) , Montfavet , France
19 GAFL, Institut National de la Recherche pour l’Agriculture, l’Alimentation et l’Environnement (INRAE) , Montfavet , France
20 Natural Resources Institute Finland , Helsinki , Finland
21 Wageningen University & Research , Wageningen , The Netherlands
Lefkowitz Elliot
Electronic publication date: 2023 Aug 16
Publication date: 2023
Volume: 11
Electronic Location ID: e15816
Received 2023 Feb 20; Accepted 2023 Jul 10
Copyright: © 2023 Rollin et al.
Copyright year: 2023
Copyright holder: Rollin et al.
License: This is an open access article distributed under the terms of the Creative Commons Attribution License, which permits unrestricted use, distribution, reproduction and adaptation in any medium and for any purpose provided that it is properly attributed. For attribution, the original author(s), title, publication source (PeerJ) and either DOI or URL of the article must be cited.
License URL: https://creativecommons.org/licenses/by/4.0/

Keywords: Bioinformatic, Genomic, Virus, Plant, Variant

Funding: COST (European Cooperation in Science and Technology) COST Action FA1407 (DIVAS) European Union’s Horizon 2020 Research and Innovation Program No 813542T (INEXTVIR) This study is the follow-up of the work on COST Action FA1407 (DIVAS), supported by COST (European Cooperation in Science and Technology). This was supported by European Union’s Horizon 2020 research and innovation program under the Marie Skłodowska-Curie grant agreement No 813542T (INEXTVIR). The funders had no role in study design, data collection and analysis, decision to publish, or preparation of the manuscript.

==============================
Recent developments in high-throughput sequencing (HTS) technologies and bioinformatics have drastically changed research in virology, especially for virus discovery. Indeed, proper monitoring of the viral population requires information on the different isolates circulating in the studied area. For this purpose, HTS has greatly facilitated the sequencing of new genomes of detected viruses and their comparison. However, bioinformatics analyses allowing reconstruction of genome sequences and detection of single nucleotide polymorphisms (SNPs) can potentially create bias and has not been widely addressed so far. Therefore, more knowledge is required on the limitations of predicting SNPs based on HTS-generated sequence samples. To address this issue, we compared the ability of 14 plant virology laboratories, each employing a different bioinformatics pipeline, to detect 21 variants of pepino mosaic virus (PepMV) in three samples through large-scale performance testing (PT) using three artificially designed datasets. To evaluate the impact of bioinformatics analyses, they were divided into three key steps: reads pre-processing, virus-isolate identification, and variant calling. Each step was evaluated independently through an original, PT design including discussion and validation between participants at each step. Overall, this work underlines key parameters influencing SNPs detection and proposes recommendations for reliable variant calling for plant viruses. The identification of the closest reference, mapping parameters and manual validation of the detection were recognized as the most impactful analysis steps for the success of the SNPs detections. Strategies to improve the prediction of SNPs are also discussed.

Introduction

Establishing high-throughput sequencing (HTS) technologies and developing bioinformatics methods for analyzing the generated sequencing data have greatly improved knowledge of viral diversity, including plant viruses (Pappas et al., 2021). Complete or nearly complete virus genomes can be generated by de novo assembly of raw reads into longer contigs. In this process, variants can be identified, differing from the original virus (considered as the reference) by mutations, like single nucleotide polymorphisms (SNPs), i.e., insertions/deletions/substitutions (Ramesh et al., 2021). Finding individual virus variants from a mixed population via variant calling is essential for understanding viral evolution and genetic diversity. As the recent COVID-19 pandemic has shown, continuous monitoring of virus genome evolution is essential to implement societal and sanitary measures for human health (Hirabara et al., 2022). This monitoring is also key for plant viruses (Rubio, Galipienso & Ferriol, 2020), especially since it is well known that (+) RNA viruses mutate rapidly. Indeed, genome evolution due to the high mutation and recombination rates may allow RNA viruses to increase their host range and adapt to new environments (Elena, Fraile & García-Arenal, 2014; Gibbs & Weiller, 1999; Simon-Loriere & Holmes, 2011; Tromas et al., 2014; Kutnjak, Elena & Ravnikar, 2017). The generation of accurate genome is therefore becoming a key challenge for proper integration of genome data into virus epidemics monitoring. Any base modification of a genome can be decisive, so the proper detection of single nucleotide polymorphisms (SNPs) in contigs built from HTS data is a crucial step in bioinformatics pipelines (Kutnjak et al., 2015) and the identification of SNPs present at low frequencies in the sequences (and therefore absent in the consensus sequence) is important too and often overlooked.

Virus low frequency variants are important to detect as they can change the dynamic of an infection. Indeed, minor variants can change the fitness of Coxsackievirus B3 in human lung cells (Bordería et al., 2015). In several citrus varieties, Citrus tristeza virus (CTV) symptoms may be caused by low frequency variants (Černi et al., 2008). Therefore, monitoring the viral populations, including the low frequency ones, is important for plant treatment or disease prevention (Kutnjak, Elena & Ravnikar, 2017). Virus population studies often rely on consensus sequences that may hide the minor variant composition (Domingo & Perales, 2019). That is why understanding our abilities (and limits) to detect low frequency variants is important for population studies applications as shown recently on Uganda cassava brown streak virus (UCBSV) for which the analysis of minor SNPs identified three distinct haplotypes with contrasted geographical spread throughout Rwanda (Nyirakanani et al., in press).

Most of the benchmarks done on SNPs analysis focused mainly on the variant calling step (Deng et al., 2021; Barbitoff et al., 2022; Guirao-Rico & González, 2021) or dealt with specific issues like polyploidy (Clevenger et al., 2015). It was observed that mapping (alignment) seems less impactful than the variant caller (Barbitoff et al., 2022), more amplification cycles led to false positive SNPs detection (Deng et al., 2021) or to a better understanding of the frequencies impact on the variant detection (Guirao-Rico & González, 2021). For plant viruses, reference samples have been designed recently (Tamisier et al., 2021). They can be used in benchmarking studies for virus detection and genome characterization including, to a lesser extent, SNPs detection, as each sample addresses specific challenges (closely related isolates of the same species, presence of SNPs…). In eukaryotes, variants with frequencies below 5% are considered low-frequency variants and below 1% are rare-frequency variants (Guirao-Rico & González, 2021; Bomba, Walter & Soranzo, 2017); in this study, we followed these definitions.

The studies mentioned above corresponded to systematic benchmarks carried out by a single or a few research teams. Such small-scale comparisons are necessary, but they will not necessarily reflect the performance of the algorithms once the scientific community uses them. Therefore, a valuable and complementary methodology is to organize performance testing, including many laboratories that routinely use bioinformatic analyses to generate viral genomes and detect SNPs in these genomes. Performance testing of laboratory protocols by end-users has been a common approach for over a decade. It has recently been applied also for bioinformatics analyses focused on detecting plant viruses in 10 HTS samples containing 12 plant viruses that were shared between 21 participants (Massart et al., 2019) or for both laboratory and bioinformatics analyses (Gaafar et al., 2021). Performance testing by end-users has been useful in identifying the critical steps in bioinformatics analyses. For example, the detection results obtained (Massart et al., 2019) underlined the importance of the reads abundance in the detection as well as expertise in result interpretation. Performance testing is a key step toward the identification of the strength and weaknesses of protocols and contributes to the definition of reliable protocols that could be further applied by many laboratories.

Prior to this publication, preliminary performance testing was conducted on real samples (tomato artificially inoculated by an infectious clone of pepino mosaic virus (PepMV)) to monitor the laboratories’ abilities to detect SNPs (unpresented data). This preliminary evaluation delivered key outcomes such as the difficulty in validating the presence of low/rare frequency variants detected by some participants (limiting the evaluation of the performance of the bioinformatics analyses), the very high complexity of bioinformatics pipelines relying on multiple steps, and the need to focus on well-defined questions. Based on these outcomes, the decision was made to reduce the complexity by generating four artificial samples on only one well-known virus, pepino mosaic virus (PepMV). PepMV is a positive sense ssRNA classified in the species Pepino mosaic virus (genus Potexvirus, family Alphaflexiviridae) known to infect tomato, eggplant and potato plants. Because of the specificity of this study, we created our own artificial samples keeping in mind the good practices shown in the community effort on samples production for bioinformatic validation (Tamisier et al., 2021) and the usual frequency of PepMV sequences in dataset from infected tomato. In addition, the complexity of pipelines was managed by dividing the bioinformatics pipelines for SNPs detection into three fundamental steps. The three steps were decided based on our hypothesis regarding their relative importance on the final results: read quality trimming, read assembly and SNPs calling. In addition, the expert validation of SNPs detected was also independently analyzed.

We describe in this study a performance testing of bioinformatics protocols by the end users, meaning that all the steps from the reception of HTS raw data until the interpretation of SNPs predicted were evaluated. End-users corresponded to 14 plant virology laboratories with different levels of expertise in bioinformatic and variant analyses. The study reported here evaluated the sensitivity and reproducibility of variant analysis applying various bioinformatic strategies currently used in the plant virology community. Each participating laboratory received the same datasets at the same time but without information on its composition in isolates of PePMV (blind test). To precisely decipher the impact of the three above-mentioned steps, data were sent in different formats in consecutive steps: raw sequencing reads, quality-controlled reads and aligned reads where the next format was only sent after analysis of the previous was completed. Our approach is, therefore, complementary to the traditional benchmarking methodologies already carried out, as it evaluated the overall bioinformatic analysis, and not only the variant caller algorithm, to show the key points leading to the failure or success of the variant identification and the SNPs manual validation.

Materials and Methods

Origin of the sequencing data

Simulated samples from PepMV were generated using the sequencing reads simulator ART (Huang et al., 2012). ART (V2.5.8) was used for simulating paired-end reads using the built-in quality profile for HiSeq 2500. The reads were artificially generated from the chosen genome sequences, with mutations added manually to the different samples using multiple criteria (frequencies, length, quality). The detailed samples composition is shown in Table 1, in addition, all datasets are freely available (https://doi.org/10.5281/zenodo.7431632).

Table 1 Composition of samples.

Sample 1 is composed of tomato plants and the targeted virus; it comprises one isolate and nine variants (26 SNPs in total), with SNPs frequencies ranging from 60% to 0.1%. Sample 2 is a combination of PepMV, tomato plant and dark matter; it contains two isolates with nine variants (nine SNPs in total), with frequencies ranging from 60% to 0.1%. Finally, sample 3 is made of PepMV, tomato plant, dark matter and ToBRFV (tomato brown rugose fruit virus); it contains one isolate with six variants (14 SNPs in total), with frequencies ranging from 60% to 0.2%. The SNPs represented with an asterisk (*) are SNPs for which the coordinates can be different due to tandem repetition.

Sample	Organism	Reference	Variant	Number reads	Relative frequency	
Sample 1	Solanum lycopersicum	HG975518.1	All	1,600,000 (2 × 8,00,000)	80%	
PepMV	DQ000985.1	400,000 (2 × 200,000)	20%	
PepMV	DQ000985.1	A339G, A900-, ins6336T	240,480 (2 × 120,240)	60.12%	
C2668A, G319-, ins2191C	80,160 (2 × 40,080)	20.04%	
G3830C, A5439-, ins5894C	40,080 (2 × 20,040)	10.02%	
A2170T, A3769-, ins3912T	24,048 (2 × 12,024)	6.01%	
A2672C, G4090-	8,016 (2 × 4,008)	2.00%	
G3371C, T6381-, ins2557C	4,008 (2 × 2,004)	1.00%	
A5726C, G4518-, ins3885C	2,004 (2 × 1,002)	0.50%	
A2523G, A2065-, ins2942T	802 (2 × 401)	0.20%	
C5328T, G1580-, ins1334G	402 (2 × 201)	0.10%	
Sample 2	Solanum lycopersicum	HG975518.1	All	1,600,000 (2 × 800,000)	66.66%	
Dark matter	–	400,000 (2 × 200,000)	16.66%	
PepMV	DQ000985.1 & AJ606359.1	400,000 (2 × 200,000)	16.66%	
PepMV	DQ000985.1	A861C	240,480 (2 × 120,240)	60.12%	
AJ606359.1	A4918T	80,160 (2 × 40,080)	20.04%	
C339G	40,080 (2 × 20,040)	10.02%	
C3830T	24,048 (2 × 12,024)	6.01%	
T2576G	8,016 (2 × 4,008)	2.00%	
G2916C	4,008 (2 × 2,004)	1.00%	
C2523G	2,004 (2 × 1,002)	0.50%	
T1230C	802 (2 × 401)	0.20%	
G3024T	402 (2 × 201)	0.10%	
Sample 3	Solanum lycopersicum	HG975518.1	All	1,496,036 (2 × 748,018)	57.71%	
Dark matter	–	400,000 (2 × 200,000)	14.28%	
ToBRFV	MK648157.1	400,000 (2 × 200,000)	14.28%	
PepMV	DQ000985.1	400,000 (2 × 200,000)	14.28%	
PepMV	DQ000985.1	No mutation	240,480 (2 × 120,240)	60.12%	
G1983C, A4031C, A5067G	80,160 (2 × 40,080)	20.04%	
T217G, G2752A	50,502 (2 × 25,251)	12.63%	
A5489C, C4768A, ins3021A*	24,048 (2 × 12,024)	6.01%	
T1258G, T1602A*, del2363A*	4,008 (2 × 2,004)	1.00%	
A368-, ins1787T, ins3536C	802 (2 × 401)	0.20%	

Sample 1 was constructed to estimate the limit of detection of low-frequency SNPs. This sample was generated by adding one bp substitution (sometimes called single nucleotide variants (SNVs) by informatic tools), one bp deletion and one bp insertion in the sequence of PepMV, isolate CH2 with reference DQ000985.1. Nine underlying variants were generated, each containing several mutations. For instance, the first variant was generated by substituting an A for a G at position 339, deleting an A at position 900, and inserting a T after position 6,336. This sample was relatively simple, as only one isolate is present with nine (artificial) variants and was a simple combination of the targeted virus and (host) tomato plant (HG975518.1). It consists of artificial sequencing reads of 125 nucleotides long with a high-quality score (Q20:94%, Q30:88%) (see File S1). The expected difficulty in SNPs detection is on the low-frequency variants (between 60% and 0.01% frequencies) that might be below the capacity of detection of most participants.

Sample 2 was constructed to evaluate whether non-identified reads not too closely resembling the targeted virus may disturb the variant calling. This sample was generated by introducing mutations on the sequence of two isolates of PepMV—Ch2 (DQ000985.1) and Spanish LE2000 (AJ606359.1; belonging to the EU genotype). In addition, dark matter reads were created and added. Dark matter is defined as sequences of unidentified origin that are not assigned to any known taxonomic group, and may represent a problem in viral metagenomics (Krishnamurthy & Wang, 2017). This dark matter was artificially generated from a randomized sequence following the organism-specific model of Clostridioides difficile (NZ_CM000441.1) using the “random-seq” tool of the RSAT platform (November 2019) (Nguyen et al., 2018). Overall, this sample was composed of the targeted virus, tomato plant, and dark matter with reads that were 125 nucleotides long and presented a high quality (Q20:94%, Q30:89%) (see File S1). The difficulty of variant detection for this sample was supposed to be identifying the two isolates and detecting the low-frequency variants despite the background noise of the dark matter.

Finally, sample 3 was constructed to examine whether adding sequence noise/dark matter and reads representing a related virus would impact variant detection. Only one isolate was used (DQ000985.1), for which the most frequent (60%) variant is identical to the reference. Variant three was generated at 6% frequency with the reverse of the positive sense RNA of PepMV, therefore the prediction (T925G, G1646T, ins3392T* in forward, and reverse A5489C, C4768A, ins3021A*) can be considered as correct. On variants 3 and 4, there was a mutation in homopolymers (same nucleotide repetition) area, meaning that the coordinate predicted can change a little. As an example, the deletion del2363A* is in a region on which there are four ‘A’ in a row (2,363–2,366), meaning the SNPs predictor cannot know which one of the four ‘A’ was deleted; therefore, any predicted ‘A’ deletion within those coordinates was considered as correct. In addition to the PepMV, tomato, dark matter (same as sample 2), tomato brown rugose fruit virus (ToBRFV, MK648157.1) were added (see Table 1). All reads on this sample are 250 nucleotides long and present a medium quality (Q20:88%, Q30:59%) (see File S1). The background noise (dark matter and ToBRFV) is more important than sample 2 and might have a bigger impact on the variant calling ability (especially with ToBRFV reads closer to PepMV than the dark matter). The potential impact of the read quality difference could be compared to sample 2. The longer reads coupled with the lower quality should allow more diversity between participants’ result of the cleaning of the reads.

Organization of the performance testing

In total, 14 laboratories from eight different countries participated in the performance testing reported here with the coordination of Liège University (Belgium). Each participant was free to choose the algorithms and parameters for each of the three defined steps of the bioinformatics analysis. Importantly, the participants carried out the process step by step, as described in Fig. 1.

Figure 1 Analysis steps.

The test was organised into three steps for each sample. (I) Pre-processing of reads, three datasets were selected from the results to be performed by every lab for the second step. (II) Identification of target virus reference, three mapping files resulting were selected for the last step. (III) Variant calling, the way SNPs predictor tools were used, and manual validation of detection were compared.

In order to prevent too much diversity in the approaches used, a selection of intermediate process results was used as common datasets from which each lab had to start the next step. The overall idea was to allow the comparison of method/laboratory performance as each participant used the same input files for each step. The information on the bioinformatics analyses (anonymized by replacing laboratories name by letters) carried out for all steps by the 14 participating laboratories is available in File S2.

I. Pre-processing step

The participants received the three designed datasets in blind. They might, in this step, perform actions to select reads to keep for further analysis, like pairing, merging, trimming, or nothing at all. The results they were asked to send corresponded to a FASTQ file for each dataset containing sequences considered ready by the participant for the following steps. In addition, the bioinformatics analysis carried out had to be reported (algorithm and parameters used).

To compare the received FASTQ files (14 files from each sample = 42 files), the PT coordinator used several criteria (see next chapter). From each original dataset, three files (referred as B, C, and D) showing the most divergent conditions of pre-processing reads (relax cleaning, average cleaning and strict cleaning) were selected for the next step. In addition, the initial files referred to as A could have been used throughout all steps (optional for each participant) and were different for each participant (to try to evaluate the full pipeline impact) as different tools were used. The most pertinent criteria to define average or strict cleaning was decided according to the result observed.

II. Identification

The nine FASTQ files (three per sample) were sent to all participants in blind (they just knew the original dataset origin). In addition, the participants carried out the analysis on their own FASTQ files generated during step 1. The goal for the participants was to identify the appropriate PepMV reference sequence(s) and to align the sequencing reads on it (them) (without the need to search for other viruses). Many different methods could be applied: de novo assembly, mapping, Kmer search and dendrogram. The expected result was the mapping of reads on the correct reference(s) in bam format.

The success of this step depended on the participant’s ability to identify the correct reference(s). Further on, the coordinator analyzed the 140 bam (alignment) files received (14 laboratories × (3 samples × 3 datasets + own file)) using several criteria: number of reads mapped, the alignment quality, and coverage. From each dataset, three bam files were selected based on the criteria differences and were called E, F, and G. In total, three bam files per dataset (nine bam files in total) were selected for step 3.

III. Variant calling

The nine bam files were sent in blind to the participants (they only knew the original dataset). In addition, each participant also processed the bam file generated from their own analysis (if any).

This step involved identifying the SNPs present in each bam file by any method of their choice and confirming the detection (manual confirmation).

Criteria for data analysis at each step

For the pre-processing step, the result comparison was based on classical reads statistics: number, quality and length. In addition, the tools and parameters used by the participants (pairing, merging, adapter removing, quality trimming, duplicate trimming) were gathered. The procedure can change the expected output significantly if duplicates are removed and reads are paired and merged, generating a lower number but sometimes longer (when merged) reads. The statistics were obtained from Geneious Prime (2020.0.5; Biomatters, Auckland, New Zealand), and the procedure (for all steps) is available in File S2.

In the second step, the mapping result files were compared using Geneious Prime (2020.0.5; Biomatters, Auckland, New Zealand) to extract the following statistics: reference used, number of reads mapped, mean coverage, coverage range, identical sites) while keeping track of tools used to examine how they affected the observed results.

Finally, for SNPs calling, the comparison was more challenging as there is no standard way to compare SNPs detection. Due to the high number of files obtained from different sources (n = 140), the analysis of the data files was automated by a custom script. Furthermore, the use of terminology in fields of the results files was checked and harmonized between laboratories. As an example, the “Manual validation field” could contain several different words for the same meaning (“YES”, “yes”, “ok”, etc.), that needed to be replaced with only one term per meaning.

The inferred SNPs were compared to the expected SNPs using three parameters: (i) mutation type (only SNPs were considered), (ii) position of the SNPs, and (iii) reference/allele correspondence (A/T C/-, etc.). The position of each predicted SNPs was compared to the expected position of the mutation in each sample. A tolerance of +/−1 was accepted for all positions as some predictors may start to count with 0 or 1. In addition, some positions were less precise, as a mutation in homopolymer regions can be reported correctly at several coordinates. Finally, for the third variant of sample 3, the mutation and reads were given from the reverse strand, and all positions for reverse and forward strands were checked. The comparison methodology applied for the analysis of the results is available as a Python notebook (v3.7) here: https://github.com/johrollin/jupyter_variant_calling

Criteria for evaluating SNPs detection

To be considered “correct”, the detection of SNPs had to pass the three parameters described above. To obtain the “correctness” analysis, we made two confusion matrices: the first one is based on the software detection (S). In contrast, the second one uses the manual validation (MV) performed.

Confusion matrices are layouts that allow the performance monitoring of predictions. On the software part, we considered as True Positive (TPs) all the expected SNPs that were predicted (regardless of their validation status by the participant), the False Positive (FPs) as all the wrong detections of SNPs, the False Negative (FNs) as the number of expected SNPs that were not predicted. Finally, the True Negative (TNs) was not calculated as it did not make sense here (the number of non-predicted SNPs can not be known). We also calculated the True Positive and False negative rates, which correspond to the percentage of expected SNPs found (TP) or missed (FN) among the total number of expected SNPs for the sample (see Table 1).

On the manual validation matrix, a True Positive (TPmv) was an expected SNPs predicted with a positive validation status. A False Positive (FPmv) corresponded to the wrong detection of SNPs that were positively validated. A False Negative (FNmv) was an expected SNPs validated as negative, and, finally, a True Negative (TNmv) represented a wrongly predicted SNPs that was negatively validated (rejected by manual validation). We also calculated the True Positive and False negative rates, which corresponded to the percentage of expected SNPs found and positively validated (TP) or negatively validated (FN) among the total number of expected SNPs predicted by the software for the sample.

Results

Result impacting decision on protocol

Step 1: reads pre-processing

The objective of the pre-processing step was to evaluate the difference in the read-cleaning strategies of participants and to further evaluate their impact on the next steps of the analysis. We obtained several FASTQ files from each participant (65 files in total); the statistics on the reads retained after this step files were studied and are available in File S3.

The data transformations carried out by the participants, in various combinations were: Pairing: associate the two reads of the same pair together; the modification made on one of the reads may impact the second one.

Merging: merge the two paired reads to obtain one longer read.

Adapter trimming: removing small (Illumina) adapters that may be on reads because they can interfere with downstream analyses.

Quality trimming: remove nucleotides (or read) based on their base calling quality score.

De-duplication: remove identical reads if several are present.

Not all of these data transformations were performed by all laboratories, and the order in which they were performed was also different. These transformations and their order are important because they can impact the number of reads, their average length and the quality of the remaining reads, which were the metric for the dataset selection for the following step. The individual effect of each transformation on reads is known (Lebas et al., 2022). In our case, it is their combination that is interesting.

The results were analyzed according to two axes. First, the consistency in read number, quality, and length between the original file and the cleaned ones were checked between the three samples. In the second axis, the differences observed in applied methodology and obtained results were compared between laboratories.

An obvious trend is that a higher average base quality is obtained when more nucleotides are removed. Merging was performed by three participants and was identified as a critical step in comparing the different datasets since it significantly changed the statistics. Therefore, those datasets were analyzed separately from the non-merged ones. When merging, the quality of the remaining read is higher, but the number of remaining nucleotides is lower. That is expected since the overlapping part of a pair of reads is used for merging. On the non-merged resulting FASTQ files, we saw that similar combination of methods provided similar results, with the slight difference explained by the variation of parameter/version of used tools. No unexpected results were obtained on these steps when analyzed alone (before comparing to other step).

To monitor the eventual effect of the pre-processing on the overall analysis, a selection of datasets to be used by all labs for the next step was performed. The criteria for the selection were discussed and agreed among participants. A combination of metrics corresponding to read number and length (impactful on coverage ability which is believed to be very significant for later variant calling), and quality score was used to select the datasets that are described in Table 2. The Illumina adapter effect on the trimming was not evaluated as they are not generated by the read simulator tool.

Table 2 summary of quality metrics of the nine selected datasets for the following steps.

The metrics number of reads (Read NB), read length range (minimum to maximum read length), mean (read) length and the number of nucleotides (NB nucleotides) were used to monitor the remaining amount of informative data remaining. Mean confidence (based on confidence scores provided by the base calling program) and the quality score metrics were used to assess the quality/confidence of the remaining data. Quality scores are an estimation of the probability of a base (call) being wrong, Q20 represent 1 in 100 error rates, Q30 is 1 in 1,000, Q40 1 in 10,000, meaning that a higher Q score indicates a smaller probability of error, but keeping only the high quality reads mean less informative data kept for further analysis.

Condition	Sample	Reads NB	Read length range	Mean length	Mean confidence	Q20	Q30	Q40	NB nucleotides	
B	1	979,286	30–230	131	38	99%	95%	71%	128,377,190	
B	2	1,031,979	30–230	133	38	99%	95%	68%	137,398,669	
B	3	1,348,006	68–400	400	31	94%	73%	19%	538,687,668	
C	1	1,246,836	20–125	118	37	99%	95%	29%	146,452,744	
C	2	1,644,810	20–125	119	37	98%	94%	21%	195,589,523	
C	3	898,660	20–250	113	30	96%	69%	0%	101,325,451	
D	1	1,969,770	48–125	116	36	96%	90%	18%	228,638,480	
D	2	2,369,778	45–125	118	36	96%	91%	15%	278,612,958	
D	3	2,660,434	50–250	187	29	93%	67%	0%	497,181,717	

Condition B was kept for step 2 to include one file with merged reads as its result files were more constant in terms of high quality and read number (operations used to generate each dataset are available in File S2). It also presented a higher number of merged reads than the remaining unmerged reads (File S3). This dataset represented high-quality reads mostly merged. Conditions C had better quality confidence than average, even if it provided fewer reads than the others, which may play a key role in the last step. Condition D is similar to the majority of other datasets by most metrics. The fact that it did not have any outlier metrics made it a good candidate for the next step as a standard read cleaning case representative.

Step 2: Reference identification and read mapping

The participants aimed to identify the correct reference(s) and to provide a mapping file for further variant calling. Information about those results was extracted and detailed in File S4. The key result of this step was reference identification. The participants used a variation of assemblers (Geneious, CLC Genomics Workbench, Tadpole, Velvet, Spades) and/or alignment (BLAST or mapping) approaches. Alternative strategies of de novo assembly and mapping to all PepMV references (and no other virus species) were used by one participant, while de novo assembly and BLAST to a database containing all PepMV references (and no other virus species) were used by two participants. Most (8) laboratories used an approach that works without the virus prior knowledge with (de novo) assembly and alignment (BLAST) to a database of various (plant) viruses. No major difference between strategies or tools was found. In total, of the 14 participants, 11 found the correct (DQ000985.1) reference for sample 1, and 10 labs found the two correct references (DQ000985.1, AJ606359.1) for sample 2 (two labs found reference DQ000985.1 only). Finally, 11 labs found the correct reference (DQ000985.1) for sample 3. We investigated the methods used to find the reference, focusing on the case(s) where it failed and identifying four different causes for not identifying the correct reference(s): The alignment (BLAST) database used for reference identification was too small (the correct references were not present).

The database (BLAST) contained redundancies, meaning several identical reference IDs were found. This will not impact variant calling ability. Nevertheless, if several identical sequences are used for the mapping, the reads are likely to be divided between references depending on the mapping options, impacting the respecting SNPs frequencies and the confidence of their detections.

Several PepMV references are described, while only one was correct (incomplete BLAST selection criteria were applied)

Only the best BLAST result was kept, meaning that the second reference for sample 2 was not identified.

We also evaluated the resulting mapping against the selected references using statistics extracted from the bam file (read number, identical site, mean coverage, coverage standard deviation, coverage range) (see File S4). For samples 1 and 2, the metrics obtained by the laboratories were all close to the expectation. For sample 3, differences due to the impact of step 1 were observed. Table 3 displays these differences observed between conditions B, C and D.

Table 3 Mapping metrics of the chosen dataset for the following steps.

Three datasets were chosen from the participant results. The number of identical sites corresponds to identical di-nucleotide throughout all reads for a given position on the reference. Mean coverage is the average vertical coverage (also referred to as depth), corresponding to the mean number of reads covering each reference position. The coverage range is the minimum to maximum vertical coverage for each position.

Sample	Condition	Step 1 ID	Reference	NB reads	Identical site	Mean coverage	Coverage range	
1	E	B	DQ000985	364,068	1,660 (25.6%)	7,182	15–7,966	
1	E	C	DQ000985	400,001	1,569 (24.2%)	7,797	15–8,611	
1	E	D	DQ000985	400,003	1,571 (24.3%)	7,794	15–8,611	
1	F	B	DQ000985	364,145	1,467 (22.6%)	7,185	15–7,967	
1	F	C	DQ000985	400,001	1,485 (22.9%)	7,798	15–8,611	
1	F	D	DQ000985	400,007	1,484 (22.9%)	7,795	15–8,611	
1	G	B	DQ000985	366,495	162 (2.1%)	7,197	1–8,058	
1	G	C	DQ000985	400,007	1,474 (22.8%)	7,718	1–8,611	
1	G	D	DQ000985	400,026	1,449 (22.0%)	7,677	1–8,611	
2	E	B	AJ606359	151,023	3,642 (56.3%)	2,957	9–3,320	
2	E	C	AJ606359	159,530	3,603 (55.7%)	3,091	1–3,457	
2	E	D	AJ606359	159,553	3,599 (55.6%)	3,091	9–3,457	
2	F	B	AJ606359	151,053	3,455 (51.1%)	2,803	1–3,320	
2	F	C	AJ606359	159574	3,489 (52.5%)	3,006	1–3,457	
2	F	D	AJ606359	159,634	3,432 (51.6%)	3,002	1–3,457	
2	G	B	AJ606359	152,220	229 (3.0%)	2,795	1–3,339	
2	G	C	AJ606359	162,033	3,384 (50.1%)	3,016	2–3,457	
2	G	D	AJ606359	162,167	3,331 (48.7%)	2,989	1–3,457	
2	E	B	DQ000985	225,668	2,721 (42.3%)	4,450	8–4,911	
2	E	C	DQ000985	240,481	2,672 (41.5%)	4,688	9–5,220	
2	E	D	DQ000985	240,497	2,670 (41.5%)	4,687	9–5,220	
2	F	B	DQ000985	225,687	2,498 (38.4%)	4,349	1–4,911	
2	F	C	DQ000985	240,482	2,569 (39.9%)	4,625	1–5,220	
2	F	D	DQ000985	240,498	2,568 (39.5%)	4,623	1–5,220	
2	G	B	DQ000985	227,072	169 (2.3%)	4,370	1–4,972	
2	G	C	DQ000985	240,486	2,581 (39.6%)	4,544	1–5,220	
2	G	D	DQ000985	240,502	2,545 (39.0%)	4,543	1–5,220	
3	E	B	DQ000985	199,995	7 (0.1%)	12,503	15–13,588	
3	E	C	DQ000985	98,022	46 (0.7%)	1,577	3–1,814	
3	E	D	DQ000985	391,777	18 (0.3%)	11,023	14–12,033	
3	F	B	DQ000985	199,997	2 (0.0%)	12,478	15–13,593	
3	F	C	DQ000985	97,840	41 (0.6%)	1,581	3–1,819	
3	F	D	DQ000985	393,289	14 (0.2%)	11,050	14–12,079	
3	G	B	DQ000985	200,004	2 (0.0%)	11,772	2–13,593	
3	G	C	DQ000985	98,427	41 (0.6%)	1,531	1–1,821	
3	G	D	DQ000985	393,321	8 (0.1%)	10,504	1–12,079	

All conditions of samples 1 and 2 presented very homogeneous statistics. Only condition B provided fewer reads as it represented the merged condition (comparison with non-merged should not be done). On the other hand, only sample 3 showed some differences, as condition C (High quality) provided fewer reads than others. In all conditions, the number of identical sites dropped below 1%, probably because sample 3 presented lower-quality reads than the two other samples. Because the difference between statistics was not so high, we decided to select the new conditions (E, F, G) for step 3 based on the tool/parameters used.

Condition E represented the mapping results obtained using CLC genomics (QIAGEN CLC Genomics Workbench) with standard parameters (see lab B mapping parameters in File S2) used; the number of reads in that file was close to the expectation (see Table 1). Condition F is a case with Geneious standard parameters (see lab D mapping parameters in File S2) applied and read numbers also close to expectation. Finally, Condition G used Geneious with a more relaxed parameter, using the same parameter set as dataset F but allowing more mismatches (see lab Z mapping parameters in File S2) and presented divergence on identical site numbers.

The reasoning behind this selection was to test if the same SNPs would be predicted between the tools/parameters (meaning that the same reads were mapped at the same position). For example, the comparison between F and G should provide interesting information since the reads number was very close; the differences seen (if any) would be due to the very few read differences.

Step 3: SNPs calling

We obtained 9,624 SNPs detections, which were analyzed by comparing them to the expected SNPs using three parameters (mutation type, position of the SNPs, and reference/allele). All laboratories gave results using the forward strand coordinate, meaning that the reads were automatically reversed by the tool(s) used for the mapping.

Table 4 shows the expected SNPs’ results compared to the predicted ones grouped by dataset conditions combined. For example, dataset BE represents the merged reads (with high quality) mapped with CLC Genomics with standard parameters. By knowing the true positive percentages (TP%) and true positive rate, we can deduce the false positive percentages (FP%) and False negative rate (expected SNPs missed). There are no major differences between alternative pre-processing datasets (B, C, D), with 75% TP on average for all three conditions. On the alternative mapping parameter, the percentage of true positives decreases from dataset E (90% TP) to F (76% TP) and G (59% TP). On G (relaxed parameters for mapping), an overprediction of SNPs (last column) was observed and explains the lower percentage.

Table 4 Impact of the conditions combination on variant calling detection for all samples.

The condition: B (merged + high-quality control), C (intermediary quality control) and D (Standard quality control) were combined with E (CLC standard mapping parameters), F (Geneious standard mapping parameters) and G (Geneious relaxed mapping parameters) to provide statistics on variant calling detection. The True Positive percentage (TP%) correspond to the definition of TPs (Fig. 2): percentage of expected SNPs that were predicted on all detections. The true positive rate is the percentage of expected SNPs on all SNPs expected. “average NB” correspond to the average number of SNPs predicted.

Conditions	Combination	TP%	True positive rate	Average NB of SNPs predicted	
BE	Merged high QC + CLC	88%	53%	11.6	
BF	Merged high QC + Geneious	78%	54%	11.7	
BG	Merged high QC + Geneious relaxed	59%	52%	29.7	
CE	Intermediary QC + CLC	90%	52%	10.5	
CF	Intermediary QC + Geneious	77%	51%	11.2	
CG	Intermediary QC + Geneious relaxed	59%	51%	23.9	
DE	Standard QC + CLC	92%	54%	9	
DF	Standard QC + Geneious	73%	52%	12.8	
DG	Standard QC + Geneious relaxed	59%	52%	36.6	

Figure 2 Confusion matrices used in the frame of the performance testing for SNPs detection.

The intersection of the actual and predicted results is displayed with TP = correct hit, TN = correct rejection, FP = wrong detection and FN = missed hit. Values were calculated for software detection (S) and manual validation (MV).

Contrary to what was expected, sample 3, which was designed to be the most complicated sample, showed the best results (86% TP for all conditions). Sample 1 also showed very good TP (82%), while sample 2 showed only 57% TP (average TP% (all)). If we consider the true positive rate (expected SNPs found), sample 1 performs worst with 45%, sample 3 has 52%, and sample 2 shows the best true positive rate with 60%. There is more overprediction for sample 2 compared to the number of expected SNPs, which allowed us to recover more correct SNPs than for the two other samples. There were some discrepancies between conditions depending on the sample used: BG for sample 1 and condition BG, CG, and DG for sample 2 overpredicted a lot more SNPs than other conditions. This overprediction was always linked to a worse TP/FP balance, while the TP/FN balance was not different. For sample 3, performance metrics are less variable across different conditions. The overpredictions of the relaxed mapping condition (G) are observed in the cases where the number of identical sites was lower for G than other conditions (see Table 3).

Most of the participating laboratories correctly identify SNPs with relative frequencies above 1%. This is largely, because as described in File S2, most the bioinformatics pipelines are tailored to identify SNPs with frequencies above 1%. The important difference between detections for all frequencies and frequencies above 1% came from the true positive and false negative rates. Indeed, they rose when rare-frequency variants were no longer considered. First, the number of SNPs to consider is 17 (instead of 26) for sample 1, 6 (9) for sample 2 and 11 (14) for sample 3. At the same time, the true positive rate (expected SNPs predicted) rose to 65% (instead of 45%) for sample 1, 82% (60%) for sample 2 and 66% (52%) for sample 3. It indicated that a large part of the expected SNPs that were missed (false negative rate) are the ones with low frequencies. Additional analysis on SNPs frequencies showed that all laboratories obtained different frequency detections compared to the expectation for sample 2 (File 5).

Once the SNPs detection was done, the participant could manually check the prediction to confirm or reject the SNPs detection. The criteria used varied between laboratories, but common criteria can be highlighted. Almost all participants visually examined the mapping and focused their efforts on areas more prone to error (mapping low coverage or SNPs with low frequencies). Then they evaluated if the detection might have come from mismapped reads by analyzing the quality of the alignment and quality of the read sequences. More precise information on the manual validation process is available in File S2.

The average expected SNPs detection was 13 (instead of 26) for sample 1, 6 (9) for sample 2, and 8 (14) for sample 3. Interestingly, the TPmv follows the same pattern as in Table 5, where the E condition provides better results than F and G, and the overprediction of condition 1 BG and conditions 2 BG, CG and DG lower the FPmv. This is expected as most of the correct SNPs are missed by the variant calling for those conditions. As an example, in sample 2 condition BG, 25% of the SNPs were correctly found and validated (TPmv), but only 2% were correctly found and rejected by the validation (FNmv). Most of the overprediction was in TNmv (67%) as the validation rejected the wrong detections; only 7% (FPmv) of the wrong detection was validated. That means that the manual validation was very efficient in differentiating between correct and wrong detections. As evidence by the true positive and false negative rates, with true positive rates on average of 87% (sample 1), 96% (sample 2) and 86% (sample 3). The strategy of focusing on key areas of the mapping, and discarding SNPs predicted in a suspicious zone (badly aligned reads), seemed very efficient. Nevertheless, as it relies on manual (visual) examination, the quality might vary depending on the expertise of the scientist.

Table 5 Impact of the samples and condition on variant calling detection with alternative frequencies filters.

The TP% and true positive rate have the same meaning as in Table 4. To evaluate the impact of SNPs with rare-frequency (all frequencies), the performance criteria were also calculated for the SNPs with a frequency higher than 1%.

Samples	Conditions	TP% (all)	TP% (>1%)	True positive rate (all)	True positive rate (>1%)	Average NB of SNPs predicted (all)	Average NB of SNPs predicted (>1%)	
Sample 1	BE	89%	90%	48%	68%	15/26	14/17	
BF	77%	85%	46%	66%	18/26	13/17	
BG	64%	68%	44%	66%	38/26	19/17	
CE	92%	93%	47%	67%	13/26	12/17	
CF	79%	88%	43%	62%	15/26	12/17	
CG	81%	88%	43%	62%	15/26	12/17	
DE	92%	93%	47%	67%	13/26	12/17	
DF	79%	88%	43%	62%	16/26	12/17	
DG	80%	87%	43%	62%	15/26	13/17	
Sample 2	BE	87%	95%	56%	76%	6/9	5/6	
BF	74%	76%	62%	85%	8/9	7/6	
BG	30%	30%	59%	81%	42/9	38/6	
CE	87%	91%	56%	76%	6/9	5/6	
CF	64%	63%	62%	85%	10/9	10/6	
CG	15%	14%	61%	83%	48/9	47/6	
DE	94%	97%	61%	82%	6/9	5/6	
DF	50%	49%	62%	85%	15/9	15/6	
DG	14%	13%	62%	85%	58/9	56/6	
Sample 3	BE	89%	89%	54%	69%	8/14	8/6	
BF	84%	84%	54%	68%	9/14	8/6	
BG	83%	83%	54%	68%	9/14	9/6	
CE	89%	89%	53%	66%	8/14	8/6	
CF	87%	87%	49%	62%	7/14	7/6	
CG	82%	81%	49%	62%	8/14	8/6	
DE	89%	89%	54%	68%	8/14	8/6	
DF	89%	89%	51%	64%	7/14	7/6	
DG	83%	83%	51%	64%	8/14	8/6	
Average	75%	77%	52%	71%	17.30	14.07	

Discussion

The division of the overall analysis into three steps, from the reception of the reads to the SNPs detection, was very beneficial to structure the comparison between laboratories, conditions and to have checkpoints where the differences between laboratories can be traced to. It also highlighted some important points on specific steps regarding variant calling.

The relative importance of pre-processing

Our results showed that the pre-processing step was not very impactful as not much difference was observed between participants’ read files (see File S3) or sample treatment. The sample started with Q30:88% | Q20:94% (sample 1), Q30:89% | Q20:94% (sample 2) and Q30:59% | Q20:88% (sample 3). The main difference was the merged reads that provided different statistics (fewer but longer reads). Neither the alternative cleaning at a different level of read quality (conditions C and D) nor the merge (condition B) impacted the SNPs detections. No real difference in the resulting true positive rate was observed in the results. These results seemed to indicate that above a certain minimal quality limit (here Q20 > 90%), any additional quality trimming does not change the variant calling ability. No attempt on trimming with a minimum quality of 30 was made (it would have had a larger impact on sample 3). More tests with lower-quality data should be performed to confirm this hypothesis and identify the quality limit. Additionally, the effect of the presence of Illumina adaptor was not evaluated.

Importance of reference identification

The second step, including the identification of the reference genome to be used for mapping, was very important as, if it failed, the variant calling would be strongly biased. We identified the database composition as the main reason for the possible failure. Indeed, the scope of the reference database is important because if not enough representative sequences are present, the compared sequences might be too distant, and overprediction of SNPs can occur. In fact, the more reference sequences in the database, the better it is for the detection, but the analysis will take longer. In general, using the most closely related reference as possible is more beneficial, but if another (little bit less related) reference is used, this will also allow to reconstruct the genome, and investigate SNPs.

Most participants used a de novo assembly to obtain contigs (sequence fragments easier to identify). They then performed a first BLAST against a RefSeq database (including only one representative sequence) to have a first (quick) overview of the putative identity of the contigs. Then, as the contigs linked to PepMV were identified, they used the contigs (all of them or the pre-selected PepMV one) as input for a second BLAST analysis against a larger database containing all PepMV sequences to select the closest reference.

One of the laboratories used an alternative approach that also worked; first, the virus identification was performed using a similar approach (assembly and alignment) through the VirusDetect pipeline (Zheng et al., 2017). Once the PepMV virus was confirmed, all reference sequences for that virus were downloaded, and whole genome alignments with a phylogenetic tree construction were performed. Then one reference from each clade in the tree was selected, and all original reads were mapped to this group of diverse references. Once it was clear which clades of references were present, the procedure was repeated (with different combinations of references) to find the most closely related reference. A possible advantage of this approach is that mapping is faster than BLAST. Also, by analyzing the references in more detail (phylogenetically), more insight was obtained in the relationships between the publicly available sequences.

Read alignment and variant calling

The alternative mapping parameters comparison (condition E, F or G) showed some differences. Indeed, a similar true positive rate was obtained with each condition but with different levels of background noise (SNPs wrongly predicted). Condition E (mapping with CLC) and condition F (mapping with Geneious) used relatively similar parameters (described in File S2), but F gave more false positive than E. Condition G (mapping with Geneious with relax parameter) generated the most background noise with SNPs overpredictions, this was very variable between conditions. We believe that the wrong detections with relaxed parameters can be explained by erroneously adding some reads in the mapping that provided mismatches interpreted by the variant calling tool as SNPs. Even if that hypothesis seems very logical, it does not explain overpredictions in G, nor the differences seen between E and F. Table 3 showed that, between the three conditions, the metric on reads was very similar, which made the difference even more surprising. The differences between conditions corresponded to 100 reads, while the mean coverage was close to 10,000 reads (depth). This would indicate that if the wrong reads leading to wrong detections are among the 100 differential reads, we should have most of the false detections near a 1% frequency. Since Table 5 showed that most of the wrong detections are not below a 1% frequency, the explanation for the difference in mapping remains unclear. An additional analysis focusing more on mapping methods/parameter differences would be required to understand how the (minor) modifications in the read pool and placement can impact the variant calling afterwards.

Using different variant caller tools did not change the results; the only change observed was in the number of SNPs predicted when the frequency used as the limit of detection was different between laboratories. This differs from most of the variant caller benchmarks (Deng et al., 2021; Barbitoff et al., 2022; Guirao-Rico & González, 2021). But they focus mainly on the variant caller step with only sometimes considering the previous step (mapping or de novo assembly), which probably tends to highlight variant caller differences. One other explanation is also that we have a less complex sample (fewer SNPs to predict for each sample) compared to most of the benchmark samples (between 1 3274 SNPs (Deng et al., 2021)).

Impact of sample complexity on detection ability

The samples were designed with different levels of complexity. Sample 1 was simple, with tomato plant and PepMV reads present. The presence of two isolates (with 78% nucleotide identity) in sample 2 did not seem to impact the SNPs detection but rather the frequencies of the prediction compared to the expectation. Sample 3 was supposed to be the more complex as it provided lower sequencing quality (Q30:59% | Q20:88% instead of Q30: 88% | Q20:94%) and the additional viral background noise but with longer reads to compensate. Our method to build simulated samples was similar to the one in Guirao-Rico & González (2021) but it differed from Deng et al. (2021) (resequencing from known viral strain mixed). As both the read cleaning step and viral background noise were not very impactful on SNPs detection, it is not surprising that sample 3 (longer reads) provided the best result, and overall, all samples performed well.

Importance of manual validation

A manual validation step might be very important as it allows the biologist to discard SNPs that are not correct and filter the SNPs relevant for downstream analysis. In this study, most of the participants performed a visual examination of the mapping to point out the positions that could be problematic for the variant caller. This method was very efficient since, in all conditions for all samples, it allowed to discard some of the wrongly predicted SNPs. On average, the true positive rate was 90% in Table 6 (all samples, all conditions). Meaning that the validation confirms 90% of the expected SNPs that the variant caller predicted. The manual validation discarded most of the overprediction (TNmv > 67% for condition G on sample 2). This indicates that most of the over detections were on positions where the alignment was visually doubtful.

Table 6 Manual validation of the variant calling depending on the sample.

Table 6 shows the complete confusion matrix (FPmv, TPmv, FPmv, FNmv) for the manual validation of SNPs predicted by the variant caller, allowing the evaluation of the manual validation of the detection. The TPmv corresponds to the expected SNPs found and positively validated, FPmv shows the unexpected SNPs positively validated, TNmv represents unexpected SNPs negatively validated, and finally, FNmv shows expected SNPs found and negatively validated. Here, the true positive rate represents the expected SNPs found and validated and the false negative rate the expected SNPs found but not validated. In this table, the number of SNPs to consider depends on the amount predicted by the variant calling step.

Samples	Conditions	TPmv%	FPmv%	TNmv%	FNmv%	True positive rate	Average NB of SNPs predicted and with validation	
Sample 1 (average expected SNPs within prediction: 13.04)	BE	79%	5%	5%	11%	88%	13.35	
BF	66%	16%	9%	9%	89%	16.28	
BG	51%	5%	36%	8%	80%	31.5	
CE	83%	4%	1%	12%	87%	11.42	
CF	68%	15%	7%	10%	88%	14	
CG	70%	13%	7%	10%	87%	13.5	
DE	83%	4%	1%	12%	87%	11.42	
DF	68%	15%	7%	10%	88%	13.92	
DG	69%	13%	8%	10%	88%	13.57	
Sample 2 (average expected SNPs within prediction: 5.97)	BE	83%	9%	5%	4%	96%	5.21	
BF	67%	8%	22%	4%	96%	7.35	
BG	25%	7%	67%	2%	96%	39.85	
CE	83%	13%	0%	4%	96%	5.21	
CF	58%	9%	29%	4%	96%	9.14	
CG	12%	15%	72%	1%	96%	44.14	
DE	93%	3%	0%	4%	96%	5.07	
DF	43%	11%	42%	4%	96%	13.71	
DG	16%	10%	73%	1%	95%	46.14	
Sample 3 (average expected SNPs within prediction: 8.23)	BE	88%	1%	0%	11%	89%	6.85	
BF	83%	2%	5%	10%	90%	7.5	
BG	82%	1%	6%	10%	90%	7.57	
CE	85%	0%	0%	15%	85%	6.57	
CF	82%	2%	1%	15%	84%	6.42	
CG	76%	2%	8%	15%	84%	7.14	
DE	84%	0%	0%	16%	84%	6.71	
DF	83%	0%	0%	17%	83%	6.35	
DG	75%	0%	8%	17%	83%	7	
Average	69%	7%	16%	9%	90%	13.96	

The experience with SNPs detection was very variable among participants, as three out of 14 did not employ any manual confirmation. Of the 11 laboratories that performed it, that step was very beneficial for 10 of them (see File S6). A too-strict manual filter caused the failure to improve the result with the manual examination (>10% frequencies) for the remaining laboratory. In real life several aspects can make the expert curation more difficult, as too many SNPs can be predicted making manual checking a slow and long process. However, expert validation of SNPs detection is a very important step of the variant calling, and even if performed by non-experienced scientists, it remains beneficial.

Conclusion

In this study, we simulated three plant samples containing PepMV with different variants at several frequencies. Then, a performance study with 14 laboratories was carried out to highlight the key points leading to the failure or success of the SNPs detection in plant viruses. We showed that steps often considered very important (Koboldt, 2020), like pre-processing, were not that impactful when the base quality was already decent (Q30 > 88% for sample 1 and 2). The effect of the complexity of the datasets was not as conclusive because the dark matter or the viruses mix did not add the expected level of complexity. In addition, the SNPs frequency analysis only showed us that higher frequencies SNPs are easier to predict than low frequencies. The two most important factors corresponded to the strategy to identify the closest reference for mapping and the manual validation of the predicted SNPs. Finally, the mapping parameters impacted our results, with relaxed conditions performing worse. The laboratory obtained overall high TP prediction, with the same SNPs missed by most of them which show a high repeatability on the variant calling. The more experienced participants obtained improved performance thanks to their manual validation. This performance testing is useful to show a variety of strategies leading to variant calling; the community-based performed analysis shows applicable pipelines that can be used to improve end-user variant calling.

Supplemental Information

Supplemental Information 1 Datasets quality.

Click here for additional data file.

Supplemental Information 2 Pipelines.

Click here for additional data file.

Supplemental Information 3 Result step 1.

Click here for additional data file.

Supplemental Information 4 Result step 2.

Click here for additional data file.

Supplemental Information 5 SNPs frequencies.

Click here for additional data file.

Supplemental Information 6 Laboratory impact.

Click here for additional data file.

We would like to thank Gladys Rufflard for administrative support and all participating author’s institution. Special thanks to all participants of the previous COST-DIVAs action that helped this work to start.

Additional Information and Declarations

Competing Interests

Author Contributions

Data Availability

The authors declare that they have no competing interests.

Johan Rollin conceived and designed the experiments, analyzed the data, prepared figures and/or tables, authored or reviewed drafts of the article, and approved the final draft.

Rachelle Bester performed the experiments, authored or reviewed drafts of the article, and approved the final draft.

Yves Brostaux analyzed the data, authored or reviewed drafts of the article, and approved the final draft.

Kadriye Caglayan performed the experiments, authored or reviewed drafts of the article, and approved the final draft.

Kris De Jonghe performed the experiments, authored or reviewed drafts of the article, and approved the final draft.

Ales Eichmeier performed the experiments, authored or reviewed drafts of the article, and approved the final draft.

Yoika Foucart performed the experiments, authored or reviewed drafts of the article, and approved the final draft.

Annelies Haegeman performed the experiments, authored or reviewed drafts of the article, and approved the final draft.

Igor Koloniuk performed the experiments, authored or reviewed drafts of the article, and approved the final draft.

Petr Kominek performed the experiments, authored or reviewed drafts of the article, and approved the final draft.

Hans Maree performed the experiments, authored or reviewed drafts of the article, and approved the final draft.

Serkan Onder performed the experiments, authored or reviewed drafts of the article, and approved the final draft.

Susana Posada Céspedes conceived and designed the experiments, authored or reviewed drafts of the article, and approved the final draft.

Vahid Roumi performed the experiments, authored or reviewed drafts of the article, and approved the final draft.

Dana Šafářová performed the experiments, authored or reviewed drafts of the article, and approved the final draft.

Olivier Schumpp performed the experiments, authored or reviewed drafts of the article, and approved the final draft.

Cigdem Ulubas Serce performed the experiments, authored or reviewed drafts of the article, and approved the final draft.

Merike Sõmera performed the experiments, authored or reviewed drafts of the article, and approved the final draft.

Lucie Tamisier performed the experiments, authored or reviewed drafts of the article, and approved the final draft.

Eeva Vainio performed the experiments, authored or reviewed drafts of the article, and approved the final draft.

Rene AA van der Vlugt performed the experiments, authored or reviewed drafts of the article, and approved the final draft.

Sebastien Massart conceived and designed the experiments, analyzed the data, authored or reviewed drafts of the article, and approved the final draft.

The following information was supplied regarding data availability:

The script is available on GitHub and Zenodo: https://github.com/johrollin/jupyter_variant_calling.

Rollin. (2023). Detection of single nucleotide polymorphisms in virus genomes assembled from high-throughput sequencing data: large-scale performance testing of sequence analysis strategies (Version 1). Zenodo. https://doi.org/10.5281/zenodo.7948362.

The data is available at Zenodo:

Johan Rollin. (2022). Plant virus SNP prediction artificial dataset Performance Study [Data set]. Zenodo. https://doi.org/10.5281/zenodo.7431632.

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
