# Peer review of "Detection of single nucleotide polymorphisms in virus genomes assembled from high-throughput sequencing data: large-scale performance testing of sequence analysis strategies"

_PeerJ, doi:10.7717/peerj.15816_

## Round 0.1 · original submission · Minor Revisions

All three reviews provided encouraging feedback regarding your manuscript and felt it worthy of publication after a number of minor issues are addressed. Please read the reviews carefully, using them a a guide for revising your manuscript.

Reviewer 1 ·

Basic reporting

The manuscript by Rollin and colleagues addresses an important issue related to the employment of HTS technologies in virology, i.e., the prediction of single nucleotide polymorphisms, which may depend on the application of different bioinformatic pipelines.
The work is overall well organized. Figures, tables and supplementary data are descriptive and relevant, the literature is correctly cited and representative of the topic.
While the Materials and Methods, Results and Discussion sections are generally well written, I have some concerns about the language in the Introduction, which provides a contrasting impression. This section suffers from flaws which are somehow discouraging and do not give the right context to the paper. Below some changes are listed, which could contribute to improve the abstract and introduction, as well as some additional minor points.

Abstract: please rephrase:…changed research in virology, especially for virus discovery.

Line 4: Complete or nearly complete virus genomes can be generated by de novo assembly of raw reads into longer contigs.

Line 5: In this process, variants can be identified, differing from the original virus (considered as the reference) by mutations, like single nucleotide polymorphisms (SNPs), i.e., insertions/deletions/substitutions [2].

Line 8: delete pathogens'

Lines 10-11: This statement...rapidly. Please rephrase, not all plant viruses have RNA genomes. Also, it would be advisable to specify (+)RNA

Lines 12-13: Indeed, genome evolution due the high mutation and recombination rates may allow RNA viruses to increase their host range and adapt to new environments [5]-[9]. The generation of accurate genome....

Lines 14-18: Any base .... important to . Please rephrase, not clear at all and badly written.

Line 20: can change

Lines 20-25: In plant (which plant?) some (which?) of citrus tristeza virus (CTV) (no capital letter, no italics) symptoms may be cause (caused, not cause) by low frequency variants [12]. Therefore, for plant treatment (which treatment against viruses?) or disease prevention, monitoring viral population (populations?) including the low frequency one (ones?) is important [9]. Virus population studies often rely on consensus sequences that may hide the minor variant composition [13]. That is why understanding our ability (and limit) to detect low frequency variant (variants?) is important for population studies applications.

Please also note that "important" has been used five times in lines 15-25.

Line 27: It was observed that mapping (alignment)...

Line 31: for virus detection and genome characterization

Line 56: ...of only one well-known virus, pepino mosaic virus (PepMV). PepMV is a positive sense ssRNA virus classified in the species Pepino mosaic virus (genus Potexvirus, family Alphaflexiviridae) known to infect tomato, eggplant and potato plants.

Experimental design

The involvement of 14 plant virology laboratories, adopting diverse bioinformatic analyses with a uneven level of expertise, offers the possibility of a large-scale performance testing to evaluate the effect of different pipelines on SNP detection. The experimental design is well structured, the construction of the artificial samples is studied to include different frequencies of variants, and the analysis steps are well established to minimize the bias and are described in details. Whereas the target virus is clearly identified, the host plant sequence component should be better clarified. “Tomato plant” is too general and not very precise. Also “dark matter” would need a more careful description.

Please consider the following points.
Line 101 and legend of Table 1. Please specify "tomato plant", which is not informative. I suppose that it is Solanum lycopersicum chromosome ch06, complete genome.

Line 109: delete (sequence of unidentified origin). The definition of dark matter follows.

Lines 110-111: Dark matter is defined as sequences of unidentified origin that are not assigned to any known taxonomic group, and may represent a problem in viral metagenomics [23].

Line 127: In addition to PepMV, tomato plant (sequence instead of plant?), dark matter (same as sample 2), tomato brown rugose fruit (ToBRFV, MK648157.1) reads were added

Lines 132- 133: The longer reads coupled with the lower quality should allow more....

Line 169 and following: BAM or bam?

Lines 233-234: Finally...non-predicted SNPs. Please rephrase.

Validity of the findings

The results are sound and have a relevant impact on variant calling, giving tools which may have a general application and should be taken into consideration by end users. The results are discussed in depth, providing a fair analysis of all the steps adopted.

Minor changes:

Lines 286- 287: delete one “remaining”

Line 289: represents

Line 291: means

Line 368 and 370: corresponds

Line 397: Supplementary

Line 416: I suppose that one FPmv should be TNmv

Line 462: delete “you”

Line 505: …levels of complexity. Sample 1…

Line 525: …out of 14 did not employ...

Line 535: …with 14 laboratories was carried out..

Additional comments

Please note that italicization of taxa has been lost in the copy paste. Viruspecies, genus and family names should be italicized.

Reviewer 2 ·

Basic reporting

The study reported in this manuscript is within the scope of PeerJ. Authors compare the results of using SNV calling pipelines from 14 plant virology laboratories to detect viral SNV. Listed below are the review details.
1. Basic reporting
The paper compared the analytic results of using SNV calling pipelines in 14 plant virology laboratories to detect viral SNV. The paper is well organized but has some methodology gaps as well as some grammar errors listed below. Authors are encourage to go through the entire manuscript to avoid any typos and grammar errors.
(1) Title: Change ‘atrategies’ to “Strategies”

Experimental design

The research question is well defined and meaningful. However, it has some limitations.
(1) Read simulator ART does not simulate technical sequences, such as adapters. Therefore, no difference was found in trimming is understandable because it only evaluated quality trimming, not adapter trimming. However, in real sequence situation, which was not discussed in this manuscript, all sequence datasets have adapters which should be removed and benchmarked before mapping and SNV calling. Therefore, line 536-538 does not reflect the situation precisely. Please make necessary revisions.
(2) Line 309-325, and 484-487, failing to identify the correct reference in some labs might not be absolutely important in some of the cases. It is also possible to evaluate SNV calling based on alternative genome sequences. Although finding a most relevant viral genome is important in plant pathology, but SNV evaluation can also be performed using closely-related genomes, and generally does not affect the performance of the strategy because technically you can convert a SNV table of one genome to another genome, when the two genomes are closely-related and does not have unique regions. Author should evaluate the SNV calling of those labs choosing a wrong reference for their relevant results, as well.

Validity of the findings

(1) The 14 labs mostly used similar software, like CLC Genomics Workbench and Geneious, and not included popular or well-benchmarked Linux-based open-sourced bioinformatics programs. It did not evaluate the performance differences between open-sourced and close-sourced programs. It is necessary to discuss that It might be a gap between those labs and users who builds their pipelines using open-sourced software.

Additional comments

Recommendation: MAJOR REVISIONS.

Reviewer 3 ·

Basic reporting

It's quite a complex analysis and sometimes it is difficult to follow what treatments/transformations from a previous step were analyzed in the next step. I have provided some suggestions in the text on how this could be improved.

Experimental design

The design and approach are adequate, it just needs a bit more clarity for the reader to understand exactly what was analyzed in each step as described in section 1.

Validity of the findings

No comment

Additional comments

Maybe conclusions could be expanded a little as suggested in the manuscript?

Annotated reviews are not available for download in order to protect the identity of reviewers who chose to remain anonymous.

---

## Round 0.2 · accepted · Accept

All reviewers have indicated that all their concerns have been addressed in this revised manuscript. We are happy to accept your manuscript for publication.

Reviewer 1 ·

Basic reporting

I am fine with the revised version

Experimental design

I am fine with the revised version

Validity of the findings

I am fine with the revised version

Additional comments

Thank you for submitting your revised manuscript. I recommend acceptance.

Reviewer 3 ·

Basic reporting

no comment

Experimental design

No comment

Validity of the findings

No comment

Additional comments

Authors have responded satisfactorily to reviewers comments.